# The Learnability of In-Context Learning

**Noam Wies, Yoav Levine & Amnon Shashua**
The Hebrew University of Jerusalem
`{noam.wies,yoav.levine,shashua}@cs.huji.ac.il`

## Abstract

In-context learning is a surprising and important phenomenon that emerged when modern language models were scaled to billions of learned parameters. Without modifying a large language model's weights, it can be tuned to perform various downstream natural language tasks simply by including concatenated training examples of these tasks in its input. Though disruptive for many practical applications of large language models, this emergent learning paradigm is not well understood from a theoretical perspective. In this paper, we propose a first-of-its-kind PAC based framework for in-context learnability, and use it to provide the first finite sample complexity results for the in-context learning setup. Our framework includes an initial pretraining phase, which fits a function to the pretraining distribution, and then a second in-context learning phase, which keeps this function constant and concatenates training examples of the downstream task in its input. We use our framework in order to prove that, under mild assumptions, when the pretraining distribution is a mixture of latent tasks (a model often considered for natural language pretraining), these tasks can be *efficiently* learned via in-context learning, even though the model's weights are unchanged and the input significantly diverges from the pretraining distribution. Our theoretical analysis reveals that in this setting, in-context learning is more about identifying the task than about learning it, a result which is in line with a series of recent empirical findings. We hope that the in-context learnability framework presented in this paper will facilitate future progress towards a deeper understanding of this important new learning paradigm.

## 1 Introduction

The practice of pretraining language models (LMs) over massive general purpose text corpora has revolutionized natural language processing in recent years (Radford et al., 2019; Devlin et al., 2019; Brown et al., 2020). After an LM has been pretrained, the common approach for applying it to a specific downstream natural language task is to further train it on task-specific data, in a procedure referred to as fine-tuning (Howard & Ruder, 2018; Radford et al., 2018; Devlin et al., 2019). In their influential GPT3 paper, Brown et al. (2020) showed that a non-trivial alternative to fine-tuning emerges when the LM is large enough: an LM can be specialized to a downstream natural language task by simply receiving in its input a string composed of concatenated training examples of this task. Importantly, while the LM's weights are *unchanged* in this procedure, some form of learning evidently takes place; the performance on the downstream task was shown to significantly improve with the number of concatenated training examples, for a disparate variety of natural language tasks.

This phenomenon, referred to as *in-context learning*, has had a profound practical impact on the applicability of large LMs: one does not need to have any access to the model weights in order to specialize the model for a certain task. Instead, a string of training examples provided even via API access to the model is enough, and often not many examples are required (Brown et al., 2020). However, despite its growing popularity in a multitude of use-cases (Bommasani et al., 2021), the reasons for the effectiveness of in-context learning in pretrained LMs are not well understood from

37th Conference on Neural Information Processing Systems (NeurIPS 2023).

a theoretical perspective. In particular, this new learning paradigm still lacks a formal definition of learning.

In this paper, we propose a PAC learning (Valiant, 1984) definition for in-context learning, along with the first finite sample-complexity learning results for in-context learning. Our results shed light on the mysterious question of why in-context learning works even though a string of concatenated input-output pairs does not resemble the natural distribution of pretraining examples. Our framework is based on an interpretation of pretraining as unsupervised multi-task learning of many natural language tasks, which dates back to the GPT2 paper (Radford et al., 2019), one of the instigators of the LM pretraining era. This view, which has since been widely adopted, was supported by experiments that revealed non-trivial zero-shot capabilities of pretrained LMs on a variety of natural language tasks. In other words, an LM pretrained only to maximize the likelihood of unlabeled text from the web, was able to perform reasonably well on a variety of downstream natural language tasks without learning from any additional training examples post pretraining, implying that many different natural tasks are directly learned during pretraining.

Following this interpretation, we prove our in-context learning results in a setting in which the pretraining distribution is a mixture of downstream tasks. Importantly, the pretraining examples are not explicitly associated with downstream tasks, but rather they are drawn from a mixture of tasks and the association between examples and tasks is latent. We prove that under mild assumptions, in-context learning is *guaranteed* to happen for a model trained by such multi-task pretraining. We show that the in-context learning mechanism of concatenating input/output pairs of a certain task allows the pretrained LM to uncover the latent task and improve its performance *without modifying its weights*. Our assumptions are quite general, namely that there is a lower bound on the probability of any mixture component and of any token, and that the downstream tasks are both distinguishable and have sufficient margin between different labels. See Section 3.1 for a formal definition of our assumptions.

The interpretation of in-context learning as uncovering tasks that were already learned during pretraining is also supported by empirical evidence. For example, both Min et al. (2022) and Lyu et al. (2023) showed that replacing the labels provided in the in-context training examples with random task labels barely affects the performance of in-context learning, implying that the in-context learning mechanism is more about identifying the task than about learning it. Similarly, both Webson & Pavlick (2022) and Lampinen et al. (2022) studied whether LMs truly understand the text of the in-context examples, and found that irrelevant text that mimic the task can have a similar effect as learning with true training examples.

Finally, the latent task inference perspective of in-context learning was also examined theoretically. Xie et al. (2022) study a language-inspired toy distribution for pretraining. Specifically, they analyze a pretraining distribution consisting of a mixture of Hidden Markov Models (HMMs), where each HMM factor corresponds to a single task. They showed that in this setting in-context learning was guaranteed, however, unlike our work, their analysis is confined only to the infinite limit of the number of in-context examples. We provide polynomial sample complexity guarantees for in-context learning, which are much more relevant to its efficiency in practice (often referred to as "few-shot"). Furthermore, Xie et al. (2022) limit their analysis only to a certain class of mixture of HMMs as pretraining distribution, while our results are for any mixture distribution under mild assumptions (see above). Lastly, their analysis assumes perfect learning of the pretraining distribution while our PAC approach captures imperfect pretraining. Beyond Xie et al. (2022), several recent papers (Akyürek et al., 2023; Dai et al., 2022; von Oswald et al., 2022) showed that from an expressivity point of view, self-attention architectures can implement a gradient descent algorithm with in-context examples. However, these works do not justify why pretraining on natural data will converge to weights that implement gradient decent, nor do they provide finite sample-complexity results for in-context learning. Overall, our presented PAC framework for in-context learning (Section 2) allows us to prove the first finite in-context learning sample-complexity results, in a quite general setting (Section 3). We hope that this framework will facilitated a further expansions of the understanding of in-context learning as a new learning paradigm.

## 2 A PAC Learnability Framework for In-Context Learning

In this section, we define a Probably Approximately Correct (PAC) learnability framework (Valiant, 1984) for in-context learning. Conceptually, we aim to adjust the PAC learnability framework, in order to capture the in-context few-shot learning capabilities of large language models (Brown et al., 2020; Wei et al., 2022a; Sanh et al., 2022; Ouyang et al., 2022).

We begin by describing the in-context learning paradigm. Let $f_\theta$ be a function fitted to a massive general purpose pretraining distribution. In order to apply $f_\theta$ for a specific downstream task, it is common to further train $f_\theta$ on pairs of the task's inputs $x_1, \ldots, x_k$ with their corresponding labels $y_1, \ldots, y_k$, in a procedure referred to as fine-tuning. In-context learning is an alternative, simpler technique, which "teaches" $f_\theta$ the task at hand by constructing a *prompt* comprised of concatenated training examples. Specifically, denote the string concatenation operator by $\oplus$, and let "$\backslash n$" be a special delimiter token. Then, the in-context learning technique is to construct a prompt $p$ by concatenating pairs of the task's inputs along with theirs labels:

$$p := x_1 \oplus y_1 \oplus \text{``}\backslash n\text{''} \oplus \cdots \oplus x_k \oplus y_k \oplus \text{``}\backslash n\text{''} \tag{1}$$

Given this prompt, the prediction is made by choosing the label $y$ that is the most likely continuation for the prefix $p \oplus x$ according to the function $f_\theta$. In other words, by predicting the label $y$ that maximizes $f_\theta (p \oplus x \oplus y)$. Notably, the prefix $p \oplus x$ does not resemble inputs that $f_\theta$ has trained on. Note that we used the newline symbol "$\backslash n$" for clarity. However, in practice the delimiter token might depend on implementation and could, for instance, be two successive newlines "$\backslash n \backslash n$" as done in the few-shot learning evaluation framework Evaluation Harness (Gao et al., 2021).

We now present our in-context learning learnability definition. Let $\mathcal{D}$ be a pretraining distribution over strings of characters from an alphabet $\Sigma$. And let $\tilde{\mathcal{D}}$ be a downstream task's distribution over pairs of strings $x \in \Sigma^\star$ and theirs corresponding labels $y \in \Sigma^\star$. We aim to define distribution-dependent learnability in the in-context learning setup, for a frozen probabilistic model $f_\theta$ that originally trained to maximize the likelihood of the pretraining distribution $\mathcal{D}$. Then, given a prompt $p \sim \tilde{\mathcal{D}}^k$ which consist of $k$ pairs of inputs and theirs corresponding labels, each independently sampled from $\tilde{\mathcal{D}}$, the model $f_\theta$ is tested on the zero-one loss of the in-context predictor:

$$L_{\text{in-context}, \tilde{D}} := \mathbb{E}_{x, y \sim \tilde{\mathcal{D}}} \left[ l_{0-1} \left( \operatorname{argmax}_{y'} f_\theta \left( p \oplus x \oplus y' \right), y \right) \right] \tag{2}$$

Since we are interested in frozen models as opposed to models that are fine-tuned, we will require that the same $f_\theta$ model achieves a low $L_{\text{in-context}, \tilde{D}}$ loss for multiple downstream tasks simultaneously. The following is our PAC learning definition for in-context learning of a model that was pretrained to maximize the likelihood of the pretraining distribution.

**Definition 1.** *Let $\mathcal{H}$ and $\tilde{\mathcal{H}}$ be hypothesis classes of pretraining distributions and downstream distributions respectively. We say that $\tilde{\mathcal{H}}$ is in-context learnable after pretraining on a pretraining distribution $\mathcal{D} \in \mathcal{H}$, if there exist a pretraining algorithm $\mathcal{A}$ and a pair of functions $m_\mathcal{H}, m_{\tilde{\mathcal{H}}} : (0,1)^2 \to \mathbb{N}$ with the following property: For every $\epsilon, \delta > 0$ and every $\tilde{\mathcal{D}} \in \tilde{\mathcal{H}}$, if the number of pretraining examples $n$ is greater than $m_\mathcal{H} (\epsilon, \delta)$, and if the number of the downstream tasks' examples $k$ is greater than $m_{\tilde{\mathcal{H}}} (\epsilon, \delta)$, then with probability of at least $1 - \delta$ (over the choice of the $n + k$ examples) the following holds:*

$$L_{\text{in-context}, \tilde{\mathcal{D}}} - \text{Bayes Error Rate} \leq \epsilon \tag{3}$$

In addition, we will say that a collection of downstream tasks $\tilde{\mathcal{H}}$ is **efficiently** in-context learnable after pretraining on $\mathcal{D}$, if both $m_\mathcal{H}$ and $m_{\tilde{\mathcal{H}}}$ in the above definition are polynomial in $\log \epsilon^{-1}, \log \delta^{-1}$ and $\log \left| \tilde{\mathcal{H}} \right|$.

In order to focus on in-context learning questions that are relevant for the natural language processing setup, and avoid complications that are not necessary for answering them, we will assume that the pretraining distribution $\mathcal{D}$ belongs to some hypothesis class $\mathcal{H}$ that is learnable by some pretraining algorithm:

**Assumption 1.** *There exists a learning algorithm $\mathcal{A}$, and a sample complexity function $m_\mathcal{D} : (0,1)^2 \to \mathbb{N}$, with the following property: for any pretraining distribution $\mathcal{D} \in \mathcal{H}$ and any $\epsilon, \delta > 0$, if the number of pretraining examples $n$ is greater than $m_\mathcal{D} (\epsilon, \delta)$, then with probability at least $1 - \delta$*

*over the pretraining examples, for any $T \geq 1$, the total variation of the conditional distributions of the $T$ 'th token is at most $\epsilon$:*

$$\max_{o_1 \dots o_T \in \Sigma} |\mathbb{P}_{\mathcal{D}} \left(o_T | o_1 \dots o_{T-1}\right) - f_\theta \left(o_T | o_1 \dots o_{T-1}\right)| < \epsilon \tag{4}$$

*where $f_\theta$ denotes the model yielded by the pretraining algorithm $\mathcal{A}$ and $f_\theta \left(o_T | o_1 \dots o_{T-1}\right)$ denotes for the conditional likelihood $\frac{f_\theta(o_1 \dots o_{T-1} o_T)}{f_\theta(o_1 \dots o_{T-1})}$. In addition, $m_{\mathcal{D}}$ is polynomial in both $\epsilon^{-1}$ and $\delta^{-1}$.*

With the above formal definition of in-context learning, we aim to shed some light on the mysterious in-context learning abilities of large LMs. Specifically, in the next section, we will shed light on the following major question: How do *frozen* pretrained models learn from prompts that do not resemble their pretraining distribution?

# 3 Guarantees on In-Context Learning

In this section, we will demonstrate the use of the above learnability framework for in-context learning and analyze a setting in which pretraining a simple language model provably leads to in-context learning capabilities. For doing so, we follow previous work and view the language modeling task as an implicit multi-task setup. Indeed, creating human-like text involves many different skills, from grammar to world knowledge, so learning a language model inevitably develops a variety of skills (Gulordava et al., 2018; Zhang & Bowman, 2018; Weber et al., 2021). The implicit unsupervised multi-task view of language modeling can be traced to the GPT2 paper (Radford et al., 2019), which revealed that pretrained LMs are capable of a wide variety of natural language tasks without the need for further training. Since then, this view has been reinforced for example by Hendrycks et al. (2021). They showed that on a diverse massive set of 57 real world text understanding tasks, the largest GPT-3 (Brown et al., 2020) language model improves over random chance by almost 13 percents points on average. Importantly, these results were obtained in a zero-shot learning setting, *i.e.* in a setting where the language model had only been pretrained to maximize the likelihood of unlabeled text. Accordingly, these results suggest that many different natural tasks are directly learned during pretraining.

We reflect the above implicit multi-task nature of language modeling by assuming that the pretraining distribution contains a latent variable that represents the task at hand. We show below that for such pretraining distributions, adding training examples to the in-context learning prompt implicitly reveals the already learned latent task. In Subsection 3.1, we present a multi-task pretraining hypothesis class, for which Subsection 3.2 shows that in-context learning reveals what task is currently being performed.

## 3.1 The analyzed latent concept hypothesis class

In this subsection, we describe the analyzed multi-task pretraining hypothesis class, as well as the corresponding downstream task hypothesis class that can be learned in-context after pretraining. To begin, we define the pretraining distribution as a mixture of multiple downstream tasks. Importantly, during pretraining the downstream task of each example is unknown, *i.e.* it is a latent variable, and thus pretraining is not equivalent to fine-tuning on the task since the model cannot simply ignore pretraining examples of irrelevant tasks. Specifically, we generate a length-$T$ pretraining example $x_1, \dots, x_T \in \Sigma$ from the pretraining distribution $\mathcal{D}$ by first sampling a concept $\phi$ from a family of concepts $\Phi$ according to a prior $\mathbb{P}(\phi)$. We then sample the tokens according to the concept's specific distribution $\mathbb{P}_\phi (x_1, \dots, x_T)$.

Moving to the downstream tasks, we will prove in-context learnability results for tasks where the underlying inputs distribution is a component $\phi$ in the pretraining mixture distribution $\mathcal{D}$. Formally, we generate a length $T$ downstream task example $x$ and the corresponding label $y$ from $\tilde{\mathcal{D}}$ by first sampling $T$ tokens $o_1, \dots, o_T$ according to the downstream task distribution $\mathbb{P}_\phi (o_1, \dots, o_T)$. Then we assemble $x$ using all tokens except the last one, and set the label to be that token *i.e.* we set that $x_t = o_t$ for any $t < T$ and that $y = o_T$. Note that in principle the concatenation of independent examples in $p$ causes a distribution drift from the pretraining distribution[1]. In this sense, the analyzed

---

[1]Formally, the distribution drift is caused by the fact that the concatenation of the "\n" after $y$ ignores the marginal probabilities of "\n".

model captures the fact that few-shot prompts are unnatural since they are not encountered during pretraining.

Now we describe assumptions about the pretraining distributions, for which we will prove our in-context learnability results. Our first assumption requires that given a delimiter token "$\backslash n$", two successive strings $s_1$ and $s_2$ that are concatenated with "$\backslash n$" are approximately independent according to $\mathbb{P}_\phi$:

**Assumption 2.** *There exists a constant $0 < c_1 \leq 1$ such that for any two strings $s_1, s_2$ in $\Sigma^\star$ and any concept $\phi$ the following holds:*

$$c_1 \leq \frac{\mathbb{P}_\phi\left(s_1 \oplus \text{``}\backslash n\text{''}\right) \cdot \mathbb{P}_\phi\left(s_2\right)}{\mathbb{P}_\phi\left(s_1 \oplus \text{``}\backslash n\text{''} \oplus s_2\right)} \leq \frac{1}{c_1} \tag{5}$$

Note that when the two successive strings $s_1$ and $s_2$ are exactly independent, the probability ratio in Equation 5 is equal to one, and the constant $c_1$ quantifies the deviation from this situation. While this assumption might sound restrictive, it is reasonable to assume that two consecutive paragraphs are not highly dependent according to the distributions in the pretraining mixture. Intuitively, we will use this assumption in order to apply concentration inequalities to the likelihood of the in-context prompt, and we will deduce that the role of the in-context prompt is to reweight the prior regarding the different mixture components. It is important to note that the approximate independence assumption for any component in the mixture does not imply approximate independence of the mixture distribution itself. Hence this reweighting is possible, since the assumption does not imply that the in-context prompt is ignored.

Beyond this approximate independence, we will also require that there exist a lower bound on the conditional probability of any single token:

**Assumption 3.** *There exist a constant $c_2 > 0$ such that for any string $s$ in $\Sigma^\star$, any character $\sigma \in \Sigma$, and any concept $\phi$ the following holds:*

$$\mathbb{P}_\phi\left(\sigma \mid s\right) > c_2 \tag{6}$$

Basically, we need this assumption in order to avoid the harm of zero likelihood due to the unnatural concatenation of input and output pairs in the prompt $p$ (where the prompt $p$ is defined in Section 2). Note that without such an assumption, in-context learning is impossible since the probability of a prompt $p$ might be zero and hence the prediction of the model in such cases becomes meaningless. Finally, we assume that the prior distribution is strictly positive. In other words, we will assume that there is a lower bound higher than zero on the likelihood of any concept appearing in the pretraining distribution.

**Assumption 4.** *There exist a constant $c_3 > 0$ such that for the prior $\mathbb{P}_\mathcal{D}\left(\phi\right)$ of any concept $\phi$, is at least $c_3$.*

Clearly, without such an assumption, the prior of the downstream task can be arbitrarily low, which means it will be nearly impossible to recognize the task. In the next subsection, we will use the above assumptions in order to provide in-context learning guarantees via the mechanism of task recognition.

## 3.2 Guarantees on In-Context Learning via Latent Concept Inference

In this subsection, we analyze the prediction of an in-context learning model in the setting described in Subsection 3.1. We show that in this setting, there is a polynomial sample complexity that guarantees in-context learning is **P**robably **A**pproximately **C**orrect. At a high level, since the pretraining is not precise and can only approximate the pretraining distribution $\mathcal{D}$ up to some error $\triangle_{\text{pretraining}} > 0$ (see Assumption 1), we will split the in-context prediction analysis into two parts. The first part will involve the simpler case of test examples $x$ and corresponding label candidates $y$ and $\tilde{y}$ for which the margin between the conditional likelihoods of the labels $\mathbb{P}_{\tilde{\mathcal{D}}}\left(y \mid x\right)$ and $\mathbb{P}_{\tilde{\mathcal{D}}}\left(\tilde{y} \mid x\right)$ is large enough. In this scenario, we show that both the deviation due to imperfect pretraining and due to imperfect task recognition is negligible. Therefore, we will conclude that such deviations do not have any impact on the loss of in-context learning. In the second scenario, when the difference between ground-truth likelihoods is small, the error rate of the Bayes optimal classifier must be high. Accordingly, even though the in-context predictor might confuse between labels, the loss in any case will be small because we only compare it to the error rate of the Bayes optimal classifier.

Starting with the first scenario, where the margin between label candidates is sufficiently large, we will prove that as more examples are added to the prompt, the in-context predictions converge to the correct label. As a preliminary step, we prove a lemma regarding the ratio of the prompt likelihoods $\mathbb{P}_\phi(p)$ (where the prompt $p$ is defined in Section 2) according to the ground-truth task versus the other mixture components. In other words, we ask how likely it is that the prompt of concatenated examples was sampled according to one of the tasks distributions versus the other tasks distributions. We will use this lemma in order to estimate the effect of the in-context prompt, on the prior regarding the different mixture components. Specifically, we denote by $\triangle_{\text{KL}}$ the minimum Kullback–Leibler divergence between the ground-truth component, and the other mixture components. Then, we prove that the ratio of the prompt probabilities according to the ground-truth components converge to zero, with rate that is exponential in both the number of in-context examples $k$, and in the minimal Kullback–Leibler divergence $\triangle_{\text{KL}}$. Intuitively, the exponential rate with regard to the number of examples comes naturally from the fact that each example in the prompt is sampled independently of the others. As a result, their effect is a multiplicative one. Additionally, the Kullback-Leibler divergence between the ground truth component and the other mixture components measures log probabilities, while we are interested in the probabilities themselves. Hence the rate is also exponential in the above Kullback-Leibler divergence. Formally, we have that:

**Lemma 1.** *Let $\mathcal{D}$ be a pretraining distribution for which assumptions 2,3 hold, and let $\phi^\star$ be a downstream task mixture component from $\mathcal{D}$ such that $\triangle_{KL} > 8 \log \frac{1}{c_1 \cdot c_2}$. Then, there exists $m_{\tilde{\mathcal{D}}} : (0,1)^2 \to \mathbb{N}$ with the following property: for any $\epsilon, \delta > 0$ and any $\phi \neq \phi^\star$, if the number of in-context examples $k$ is at least $m_{\tilde{\mathcal{D}}} (\epsilon, \delta)$, then, $\frac{\mathbb{P}_\phi(p)}{\mathbb{P}_{\phi^\star}(p)} < \epsilon$ with probability of at least $1 - \delta$ (over the choice of the $k$ in-context examples). Moreover, the above still holds when the labels in $p$ are randomly flipped, and $m_{\tilde{\mathcal{D}}}$ can be chosen such that it will be polynomial in $\log \frac{1}{\delta}, \log \frac{1}{\epsilon}, \log \frac{1}{c_1 \cdot c_2}, \frac{1}{\triangle_{KL}}$ and $T$.*

*Proof.* In essence, we prove this lemma by using the approximate independence assumption in order to apply concentration inequalities. In addition, we use Assumption 3 for bounding the distribution drift that is caused by the artificially inserted newline token. In particular, the log of the ratio of prompt probabilities according to different mixture components is concentrated around its expectation. Importantly, this expectation is equal to minus one times the Kullback–Leibler divergence between the components, plus a term that is caused by the mentioned distribution drift. Thus, we conclude that the ratio of the prompt probabilities according to the ground-truth task distribution and the other pretraining components converges to zero. Moreover, the rate of that convergence is exponential in both the number of in-context demonstrations, and the minimal Kullback–Leibler divergence between different mixture components. See full details in Section A of the appendix. □

Now we will use the above Lemma 1 to analyze the in-context predictions, namely the labels $\tilde{y}$ that maximize the likelihood of the concatenation of the in-context prompt $p$, with the example $x \oplus \tilde{y}$ according to the pretrained model $f_\theta$. Essentially, we will aim to understand when these predictions are identical to the Bayes Optimal Classifier predictions. That it, when these predictions align with the labels $y$ that maximize the likelihood of the example $x \oplus y$ as determined by downstream task distribution $\mathbb{P}_{\tilde{\mathcal{D}}}$. Since we are in the first scenario, where the margin between label candidates is large enough. We will prove that in this case, for large enough $k$, the ground truth in-context predictor also has margin that is at least half of the original margin. Hence, we will conclude that for such $k$ the ground truth in-context predictor is equal to the Bayes Optimal Classifier. Moreover, we will prove a lower bound on the margin of the in-context predictions, so the above still holds for any distribution that approximates the pretraining distribution sufficiently well.

Formally, for a test example input $x$, we define the margin $\triangle(x, y, \tilde{y})$ between two label candidates $y$ and $\tilde{y}$ as the difference between their conditional likelihoods $\mathbb{P}_{\tilde{\mathcal{D}}}(y \mid x)$ and $\mathbb{P}_{\tilde{\mathcal{D}}}(\tilde{y} \mid x)$ according to the downstream ground-truth distribution. Similarly, for a test example input $x$ and in-context prompt $p$, we define the margin $\triangle(p, x, y, \tilde{y})$ between two label candidates $y$ and $\tilde{y}$ as the difference between their conditional likelihoods $\mathbb{P}_{\mathcal{D}}(y \mid p \oplus x)$ and $\mathbb{P}_{\mathcal{D}}(\tilde{y} \mid p \oplus x)$ according to the pretraining distribution conditioned on the prompt $p$. Using these definitions, we consider triplets of test example input $x$ and two labels candidates $y, \tilde{y}$ with margin at least two times the pretraining error $\triangle_{\text{pretraining}}$ as the first scenario. In this case we will prove that for large enough $k$ the ground truth in-context predictor also has margin of at least the pretraining error $\triangle_{\text{pretraining}}$. This means that deviations arising from imperfect task recognition will be negligible in this situation. Moreover, the margin will remain larger than the pretraining error, which means the pretrained model $f_\theta$ handles this case well.

**Theorem 1.** *Let $\mathcal{D}$ and $\tilde{\mathcal{D}}$ be a pair of pretraining distribution and downstream task, for which Assumption 4 as well as the assumptions in Lemma 1 upholds. Then, there exists $m_{\tilde{\mathcal{D}}} : (0,1)^2 \to \mathbb{N}$ with the following property: for every test example $x$ and two label candidates $y, \tilde{y}$ with positive margin $\triangle(x,y,\tilde{y}) > 0$ and $\delta > 0$, if the number of in-context examples $k$ is at least $m_{\tilde{\mathcal{D}}} (\triangle(x,y,\tilde{y})/2, \delta)$, then $\triangle(p,x,y,\tilde{y}) > \triangle(x,y,\tilde{y})/2 + c_1^2 - 1$ with probability of at least $1 - \delta$ (over the choice of the $k$ in-context examples). Moreover, the above still holds when the labels in $p$ are randomly flipped[2], and $m_{\tilde{\mathcal{D}}}$ can be chosen such that it will be polynomial in $\log \frac{1}{\delta}, \log \frac{1}{\epsilon}, \log \frac{1}{c_1 \cdot c_2 \cdot c_3}, \frac{1}{\triangle_{KL}}$ and $T$.*

*Proof.* We begin by writing the difference between the ground-truth label likelihood, and the likelihood of other another label $\tilde{y}$ explicitly. Specifically, by the definition of conditional probabilities we have that:

$$\mathbb{P}_{\mathcal{D}} (y \,|\, p \oplus x) - \mathbb{P}_{\mathcal{D}} (\tilde{y} \,|\, p \oplus x) = \frac{\sum_{\phi} \mathbb{P}_{\mathcal{D}} (\phi) \left[ \mathbb{P}_{\phi} (p \oplus x \oplus y) - \mathbb{P}_{\phi} (p \oplus x \oplus \tilde{y}) \right]}{\sum_{\phi} \mathbb{P}_{\mathcal{D}} (\phi) \, \mathbb{P}_{\phi} (p \oplus x)} \tag{7}$$

Now, denote by $\phi^\star$ the mixture component of $\tilde{\mathcal{D}}$, then Assumption 2 assures us that for each component in the mixture, the textual prompt is approximately independent of the test example. So we can use Lemma 1 and get that in both the numerator and the denominator, the $\phi^\star$ term is the dominant term in the sum. Thus, we will get that the difference of the labels' likelihoods is approximately the fraction of the $\phi^\star$ terms, which is equal to the original downstream task margin. So we will conclude that the margin of the in-context predictor is at least half of the downstream task original margin. See full details in Section B of the appendix. $\square$

We denote by $\triangle_{\tilde{\mathcal{D}}}$ the minimal margin of the Bayes Optimal Classifier predictions in downstream task $\tilde{\mathcal{D}}$ *i.e.* the minimal margin between the Bayes Optimal Classifier prediction and another labels. With the above definitions and theorem, we can combine the scenario of large margins, which are preserved by the in-context predictor, with the scenario of small margins, in which the loss is minimally affected by the wrong prediction, and prove our main in-context learnability results:

**Theorem 2.** *Let $\tilde{\mathcal{H}}$ be hypothesis classes of downstream distributions, and denote by $\mathcal{D}$ a mixture distribution on $\tilde{\mathcal{H}}$ for which assumptions 1,2,3 and 4 uphold. Further assume that the margin $\triangle_{\tilde{\mathcal{D}}}$ of any downstream task $\tilde{\mathcal{D}} \in \tilde{\mathcal{H}}$ is at least $4 \cdot (1 - c_1^2)$, and the minimal Kullback–Leibler divergence between different distributions is greater than $\log \frac{1}{c_1 \cdot c_2}$. Then $\tilde{\mathcal{H}}$ is efficiently in-context learnable after pretraining on $\mathcal{D}$ (see Definition 1).*

*Proof.* Assumption 1 assures us the existence of the pretraining algorithm with a polynomial sample complexity. So we will prove the theorem with pretraining sample complexity that is derived from this algorithm, where the accuracy required from pretraining is $\frac{2}{a \cdot (1 - \alpha)}$ times the accuracy required from in-context learning (see analysis below). In addition, Theorem 1 assures us that for large enough $k$ the ground truth in-context predictions are equal to the Bayes Optimal Classifier predictions, so we will prove the theorem with downstream sample complexity that is derived from Theorem 1.

Let $\epsilon, \delta > 0$ and denote by $f_\theta$ the model that was learned during the pretraining process. We will prove that the contribution of any $x$ to the loss $L_{\text{in-context}, \tilde{D}}$ (see Equation 2) is at most $\epsilon$ and hence complete the proof. Given $x$, let $y$ be the Bayes Optimal Classifier prediction for that $x$. Then, we will split the analysis into two cases.

The first case is of examples $x, y$ such that theirs margin $\triangle(x, y, \tilde{y})$ from any alternative label candidate $\tilde{y}$ is at least $8 \cdot \triangle_{\text{pretraining}}$. In this case, Theorem 1 assures us that for large enough $k$ the ground truth in-context predictor also has margin $\triangle(p, x, y, \tilde{y})$ that is greater than $\frac{1}{2} \cdot \triangle(x, y, \tilde{y}) + c_1^2 - 1$. Now since $\triangle(x, y, \tilde{y}) \geq \triangle_{\tilde{\mathcal{D}}} > 4 \cdot (1 - c_1^2)$ we have that $\triangle(p, x, y, \tilde{y})$ is at least $2 \cdot \triangle_{\text{pretraining}}$. Thus, we conclude that in this case, the predictions of the ground truth in-context prediction and $f_\theta$ are the same. Furthermore both of them are identical to the Bayes Optimal Classifier prediction, and hence $x$ does not contribute to the loss.

---

[2]While theorem 1 provides label-insensitive results, it also predicts sensitivity to the labels in cases where the labels are the primary causes for the distinction between different tasks, as measured by the Kullback-Leibler divergence.

Moving to the second case, and denote by $\tilde{y}$ the in-context prediction of $f_\theta$. Then, the arguments from the previous paragraph assure us that $\mathbb{P}_{\phi^\star}(y \,|\, x) - \mathbb{P}_{\phi^\star}(\tilde{y} \,|\, x) < 8 \cdot \triangle_{\text{pretraining}}$, since otherwise we will get that $f_\theta(y \,|\, x) > f_\theta(\tilde{y} \,|\, x)$. Finally, we will choose that $\triangle_{\text{pretraining}} = \frac{\epsilon}{8}$, and hence get that also in the second case, the contribution of $x$ to the loss is less than $\epsilon$. □

## 4 Related Work

Since the Probably Approximately Correct (PAC) learning framework was introduced in Valiant (1984), a rich line of works has extended the framework to distribution-dependent bounds (Benedek & Itai, 1991; Vayatis & Azencott, 1999; Sabato et al., 2013) *inter alia*. These works relax PAC's adversarial requirement of generalization for all input distributions, and only consider distributions that satisfy certain statistical properties. Consequently, these papers provide more realistic sample complexity bounds. In this paper we present a framework for in-context learning learnability that uses self-supervised pretraining to assist in solving downstream tasks. In this framework, we follow the above line of distribution-specific works with the distinct feature of a self-supervised pretraining phase, and of learning with frozen models through their input context. The advantages of such pretraining have been studied from a theoretical perspective before. For example, Saunshi et al. (2021) and Wei et al. (2021) demonstrate that language modeling can benefit downstream tasks either by prompt tuning or head tuning. However, unlike our work, in their analysis some weights are learned. In contrast, we focus on frozen models that can learn only from their input context, which includes the concatenation of few inputs and outputs pairs.

Returning to in-context learning, a recent line of work study learning paradigms that are based on concatenating several inputs into a single input context. For example, Garg et al. (2022) show that a Transformer architecture is able to discover in-context learning algorithms for simple function classes, such as two-layer neural networks, and decision trees. Similarly, Akyürek et al. (2023) show that a Transformer architecture is able to discover efficient Bayes optimal least square learning algorithms, and Laskin et al. (2022) show that a Transformer architecture is able to discover efficient in-context reinforcement learning algorithms. Additionally, Levine et al. (2022) studied the inductive bias of in-context learning, and proved that Transformer based language models can model much stronger dependencies between text segments that appeared in the same training example. Finally, Li et al. (2023) proved a multi-task generalization bound for in-context learning with the Transformer architecture. However, unlike our work, the pretraining distribution in all of the above works designed to include input context that includes few-shot demonstrations explicitly. As a consequence, these methods do not answer the mysterious question of how frozen pretrained models can learn from in-context prompts that do not resemble their pretraining distribution.

Another recent line of work investigates possible mechanisms that enable in-context learning. For example, Olsson et al. (2022) provide indirect evidence that *induction heads* might constitute the mechanism for in-context learning in large transformer models. In addition, several recent papers (Akyürek et al., 2023; Dai et al., 2022; von Oswald et al., 2022) showed that from an expressivity point of view, self-attention architectures can implement a gradient descent algorithm with in-context examples. Finally, Chan et al. (2022) provided empirical evidence that some data distributional properties encourage in-context learning even when the frozen models do not see prompts during pretraining. However, unlike our work, all of the above works does not provide theoretical guarantees of in-context learnability.

Finally, Xie et al. (2022) are the closest to our work. They studied a language-inspired toy distribution for pretraining and analyzed a pretraining distribution consisting of a mixture of Hidden Markov Models (HMMs). Unlike their results, our results are applicable to any mixture distribution that meets our mild assumption. In addition, unlike our polynomial sample complexity guarantees, their analysis guarantees in-context learning only for an infinite number of in-context examples. Lastly, their analysis assumes perfect pretraining distribution learning, but our approach captures imperfect pretraining.

## 5 Conclusion

The discovery of in-context learning in large LMs, made by Brown et al. (2020), was surprising to many in our field. A model that was pretrained to maximize the likelihood of natural text was able

to make use of *concatenated* training examples of downstream natural language tasks—inputs that do not resemble its pretraining distribution, and moreover these inputs improved the model's ability to perform the task. Our theoretical results, based on a common latent multitask framework for the pretraining phase, shed light on the above surprising mysteries. With our PAC-based framework, we were able to provide sample complexity guarantees for in-context learning in such pretrained models, which are not only the first finite sample complexity results for this framework but they also indicate efficient (polynomial) in-context learning, which reflect the behavior of this setting in practice.

We hope that our framework can be used to deepen the understanding of the in-context learning phenomenon. In particular, we mark the connection between model size and the in-context learning efficiency as an interesting open question (Wei et al., 2022b). Additionally, in-context learning has shown to be capable of learning new tasks not included in the pre-training distribution (Wei et al., 2023). The extensions of our results to such situations, as well as input-dependent bounds that capture the sensitivity of few-shot in-context learning to the order of the few-shot examples is an interesting open questions.

**Limitations:**   While we assume that the pre-training distribution perfectly matches a mixture of downstream tasks, we acknowledge that the pre-training data in real-world LLMs is often noisy and imperfect. Our results represent an idealized scenario with perfect pre-training data. Extending our analysis to account for limitations of real-world pre-training data remains an important direction for future work.

## Acknowledgments and Disclosure of Funding

The authors would like to thank Yotam Wolf and Oshri Avnery for their assistance and advice. This research was supported by the ERC (European Research Council) and the ISF (Israel Science Foundation).

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

# A  Proof of Lemma 1

We begin by explicitly writing $\mathbb{P}_\phi(p)$. Specifically, Assumption 2 assure us that each in-context example $x_i \oplus y_i \oplus "\backslash n"$ is approximately independent from the other in-context examples:

$$c_1^k \leq \frac{\prod_{i=1}^k \mathbb{P}_\phi\left(x_i \oplus y_i \oplus "\backslash n"\right)}{\mathbb{P}_\phi(p)} \leq c_1^{-k} \tag{8}$$

In addition, Assumption 3 bounds the distribution drift that caused by the artificial new line token:

$$\mathbb{P}_\phi\left(x_i \oplus y_i\right) \leq \mathbb{P}_\phi\left(x_i \oplus y_i \oplus "\backslash n"\right) \leq \mathbb{P}_\phi\left(x_i \oplus y_i\right) \cdot c_2 \tag{9}$$

Similarly, we can use Assumption 3 again to bounds also the distribution drift that caused by potential label flipping:

$$c_2 < \frac{\mathbb{P}_\phi\left(x_i \oplus \tilde{y}_i\right)}{\mathbb{P}_\phi\left(x_i \oplus y_i\right)} < \frac{1}{c_2} \tag{10}$$

where $\tilde{y}_i$ denote the "corrected" label. So overall, we got that:

$$\left(c_1 c_2^2\right)^k < \frac{\prod_{i=1}^k \mathbb{P}_\phi\left(x_i \oplus \tilde{y}_i\right)}{\mathbb{P}_\phi(p)} < \left(c_1 c_2^2\right)^{-k} \tag{11}$$

Now, we can write the ratio of the prompt probabilities according to the ground-truth component and the other mixture components explicitly and get that:

$$\log \frac{\mathbb{P}_\phi(p)}{\mathbb{P}_{\phi^\star}(p)} \leq \sum_{i=1}^k \log \frac{\mathbb{P}_\phi\left(x_i \oplus \tilde{y}_i\right)}{\mathbb{P}_{\phi^\star}\left(x_i \oplus \tilde{y}_i\right)} + 4k \log \frac{1}{c_1 c_2} \tag{12}$$

Intuitively, as each term in the sum is independent of the other, and equals (in expectation) to the Kullback–Leibler divergence between $\phi^\star$ and $\phi$:

$$\mathbb{E}_p\left[\frac{1}{k}\sum_{i=1}^k \log \frac{\mathbb{P}_\phi\left(x_i \oplus \tilde{y}_i\right)}{\mathbb{P}_{\phi^\star}\left(x_i \oplus \tilde{y}_i\right)}\right] = -\text{KL}\left(\mathbb{P}_{\phi^\star}, \mathbb{P}_\phi\right) \tag{13}$$

Equation 12 assure us that as long as the ground truth downstream task component $\phi^\star$ distinguish enough from the rest of the mixture components, $\frac{\mathbb{P}_\phi(p)}{\mathbb{P}_{\phi^\star}(p)}$ converge to zero. Formally, Assumption 3 assure us that

$$\left|\log \frac{\mathbb{P}_\phi\left(x_i \oplus \tilde{y}_i\right)}{\mathbb{P}_{\phi^\star}\left(x_i \oplus \tilde{y}_i\right)}\right| \leq T \log \frac{1}{c_2} \tag{14}$$

for any $i, p, \phi, \phi^\star$. Hence, we can use Hoeffding's inequality (Hoeffding, 1994) to estimate the deviation of this variable from its expectation (see Equation 13). More specifically, together with Equation 12 we get that with probability of at least $1 - \delta$ over the choice of the $k$ examples in $p$ we have that:

$$\frac{\mathbb{P}_\phi(p)}{\mathbb{P}_{\phi^\star}(p)} < \exp\left(-\frac{k}{2}\left(\text{KL}\left(\mathbb{P}_{\phi^\star}, \mathbb{P}_\phi\right) - 8\log\frac{1}{c_1 c_2}\right)\right) \tag{15}$$

for $k > \frac{\left(\log \frac{1}{\delta}\right)\left(16T^2\right)\left(\log^2 \frac{1}{c_2}\right)}{\text{KL}\left(\mathbb{P}_{\phi^\star}, \mathbb{P}_\phi\right)^2}$. Finally, Lemma 1 follows by choosing that $m_{\tilde{\mathcal{D}}}$ is the maximum between $k > \frac{\left(\log \frac{1}{\delta}\right)\left(16T^2\right)\left(\log^2 \frac{1}{c_2}\right)}{\text{KL}\left(\mathbb{P}_{\phi^\star}, \mathbb{P}_\phi\right)^2}$ and $\frac{2\log\frac{1}{\epsilon}}{\triangle_{\text{KL}} - 8\log\frac{1}{c_1 c_2}}$, since the first term take care of the $1 - \delta$ probability, and the second term term take care of the $\epsilon$-approximation.

# B  Proof of Theorem 1

We begin by writing the difference labels likelihood explicitly. Specifically, by the definition of conditional probabilities we have that:

$$\mathbb{P}_\mathcal{D}\left(y \mid p \oplus x\right) - \mathbb{P}_\mathcal{D}\left(\tilde{y} \mid p \oplus x\right) = \frac{\sum_\phi \mathbb{P}_\mathcal{D}(\phi)\left[\mathbb{P}_\phi\left(p \oplus x \oplus y\right) - \mathbb{P}_\phi\left(p \oplus x \oplus \tilde{y}\right)\right]}{\sum_\phi \mathbb{P}_\mathcal{D}(\phi)\,\mathbb{P}_\phi\left(p \oplus x\right)} \tag{16}$$

Now, Assumption 2 assure us that for each component $\phi$ in the mixture the prompt $p$ is approximately independent from the test example:

$$c_1 \cdot \mathbb{P}_\phi (p) \cdot \mathbb{P}_\phi (x \oplus y) \leq \mathbb{P}_\phi (p \oplus x \oplus y) \leq \frac{1}{c_1} \cdot \mathbb{P}_\phi (p) \cdot \mathbb{P}_\phi (x \oplus y) \tag{17}$$

$$\mathbb{P}_\phi (p \oplus x) \leq \frac{1}{c_1} \cdot \mathbb{P}_\phi (p) \cdot \mathbb{P}_\phi (x) \tag{18}$$

So substituting this inequalities in Equation 16 give us that:

$$\mathbb{P}_\mathcal{D} (y \,|\, p \oplus x) - \mathbb{P}_\mathcal{D} (\tilde{y} \,|\, p \oplus x) \geq \frac{\sum_\phi \mathbb{P}_\mathcal{D} (\phi) \cdot \mathbb{P}_\phi (p) \cdot \left[ c_1^2 \cdot \mathbb{P}_\phi (x \oplus y) - \frac{1}{c_1^2} \cdot \mathbb{P} (x \oplus \tilde{y}) \right]}{\sum_\phi \mathbb{P}_\mathcal{D} (\phi) \cdot \mathbb{P}_\phi (p) \cdot \mathbb{P}_\phi (x)} \tag{19}$$

Now, we will separate the sums in the denominator and the numerator to term that contain the mixture component $\phi^\star$ that corresponds to $\tilde{\mathcal{D}}$, and term that contain the other mixture components. So for clarity we will denote by $A, B, C, D$ the different sums:

$$A := \mathbb{P}_\mathcal{D} (\phi^\star) \cdot \mathbb{P}_{\phi^\star} (p) \cdot \left[ c_1^2 \cdot \mathbb{P}_{\phi^\star} (x \oplus y) - \frac{1}{c_1^2} \cdot \mathbb{P}_{\phi^\star} (x \oplus \tilde{y}) \right] \tag{20}$$

$$B := \sum_{\phi \neq \phi^\star} \mathbb{P}_\mathcal{D} (\phi) \cdot \mathbb{P}_\phi (p) \cdot \left[ c_1^2 \cdot \mathbb{P}_\phi (x \oplus y) - \frac{1}{c_1^2} \cdot \mathbb{P}_\phi (x \oplus \tilde{y}) \right] \tag{21}$$

$$C := \mathbb{P}_\mathcal{D} (\phi^\star) \cdot \mathbb{P}_{\phi^\star} (p) \cdot \mathbb{P}_{\phi^\star} (x) \tag{22}$$

$$D := \sum_{\phi \neq \phi^\star} \mathbb{P}_\mathcal{D} (\phi) \cdot \mathbb{P}_\phi (p) \cdot \mathbb{P}_\phi (x) \tag{23}$$

And get that:

$$\mathbb{P}_\mathcal{D} (y \,|\, p \oplus x) - \mathbb{P}_\mathcal{D} (\tilde{y} \,|\, p \oplus x) \geq \frac{A}{C + D} + \frac{B}{C + D} \tag{24}$$

Now we know that both $C$ and $D$ are non-negative and hence that:

$$\left| \frac{A}{C + D} - \frac{A}{C} \right| = \left| \frac{AD}{C^2 + DC} \right| \leq \left| \frac{A}{C} \right| \cdot \left| \frac{D}{C} \right| \tag{25}$$

$$\left| \frac{B}{C + D} - \frac{B}{C} \right| = \left| \frac{BD}{C^2 + DC} \right| \leq \left| \frac{B}{C} \right| \cdot \left| \frac{D}{C} \right| \tag{26}$$

And by definition we have that:

$$\frac{A}{C} = \triangle (x, y, \tilde{y}) + \left( c_1^2 - 1 \right) \mathbb{P}_\phi (y \,|\, x) + \left( \frac{1}{c_1^2} - 1 \right) \cdot \mathbb{P}_\phi (\tilde{y} \,|\, x) > \triangle (x, y, \tilde{y}) - 1 + c_1^2 \tag{27}$$

So it is enough to show that $\left| \frac{B}{C} \right| < \frac{1}{5} \cdot \triangle (x, y, \tilde{y})$ and that $\left| \frac{D}{C} \right| < \frac{1}{4}$, in order to prove that:

$$\mathbb{P}_\mathcal{D} (y \,|\, p \oplus x) - \mathbb{P}_\mathcal{D} (\tilde{y} \,|\, p \oplus x) > \frac{3}{4} \cdot \frac{A}{C} - \frac{5}{4} \cdot \left| \frac{B}{C} \right| > \frac{1}{2} \cdot \triangle (x, y, \tilde{y}) + c_1^2 - 1 \tag{28}$$

Starting with $\left| \frac{B}{C} \right|$, by the triangle inequality have that:

$$\left| \frac{B}{C} \right| \leq \sum_{\phi \neq \phi^\star} \frac{\mathbb{P}_\mathcal{D} (\phi)}{\mathbb{P}_\mathcal{D} (\phi^\star)} \cdot \frac{\mathbb{P}_\phi (p)}{\mathbb{P}_{\phi^\star} (p)} \cdot \frac{1}{c_1^2} \cdot \frac{1}{\mathbb{P}_{\phi^\star} (x)} \tag{29}$$

In addition, Assumption 3 assure us that $\mathbb{P}_{\phi^\star} (x) > c_2^T$, hence we conclude that:

$$\left| \frac{B}{C} \right| \leq \sum_{\phi \neq \phi^\star} \frac{\mathbb{P}_\mathcal{D} (\phi)}{\mathbb{P}_\mathcal{D} (\phi^\star)} \cdot \frac{\mathbb{P}_\phi (p)}{\mathbb{P}_{\phi^\star} (p)} \cdot \frac{1}{c_1^2 c_2^T} \tag{30}$$

Similarly, we also have that:

$$\left| \frac{D}{C} \right| \leq \sum_{\phi \neq \phi^\star} \frac{\mathbb{P}_\mathcal{D} (\phi)}{\mathbb{P}_\mathcal{D} (\phi^\star)} \cdot \frac{\mathbb{P}_\phi (p)}{\mathbb{P}_{\phi^\star} (p)} \cdot \frac{1}{c_2^T} \tag{31}$$

Now, the sum of the concepts' prior is at most one, and by Assumption 4 we have that $\mathbb{P}_{\mathcal{D}}(\phi^\star) \geq c_3$, so it is enough to show that $\frac{\mathbb{P}_\phi(p)}{\mathbb{P}_{\phi^\star}(p)} < \frac{\triangle(x,y,\tilde{y})}{5 \cdot c_1^{-2} \cdot c_2^{-T} \cdot c_3^{-1}} =: \tilde{\epsilon}$ for any concept $\phi \neq \phi^\star$, in order to complete the proof. Finally, Lemma 1 assure us that under the conditions of Theorem 1, there exist $\tilde{m}_{\tilde{\mathcal{D}}} : (0,1)^2 \to \mathbb{N}$ such that if the number of in-context examples $k$ is at least $\tilde{m}_{\tilde{\mathcal{D}}}(\tilde{\epsilon}, \delta)$, then with probability of at least $1 - \delta$ we get exactly that $\frac{\mathbb{P}_\phi(p)}{\mathbb{P}_{\phi^\star}(p)} < \tilde{\epsilon}$. Importantly, $\tilde{m}_{\tilde{\mathcal{D}}}$ is logarithmic in the accuracy level $\tilde{\epsilon} = \frac{\triangle(x,y,\tilde{y})}{5 \cdot c_1^{-2} \cdot c_2^{-T} \cdot c_3^{-1}}$, and hence polynomial in $T$.

## C  Technical lemmas

**Lemma 2.** *Denote by $\triangle(x, y, \tilde{y})$ the ground-truth margin between two label candidates $y$ and $\tilde{y}$. And also denote by $\Delta_{\tilde{\mathcal{D}}}$ the minimal margin between the Bayes Optimal Classifier prediction and another labels, and by $\triangle(p, x, y, \tilde{y})$ the ground-truth margin between two label candidates $y$ and $\tilde{y}$ conditioned on the prompt $p$. Assuming that $\Delta_{\tilde{\mathcal{D}}}$ is greater than $1 - c_1^2$ and given the assumptions of Theorem 1, the following inequality holds:*

$$\Delta(p, x, y, \tilde{y}) > \Delta(x, y, \tilde{y})(1 - \alpha) \cdot \alpha \tag{32}$$

*where $\alpha := 1 - \sqrt{\frac{1 - c_1^2}{\Delta_{\tilde{\mathcal{D}}}}}$.*

*Proof.* By Theorem 1 we have that $\Delta(p, x, y, \tilde{y})$ is larger than $(1 - \alpha) \cdot \Delta(x, y, \tilde{y}) + c_1^2 - 1$. Now we can substitute the definition of $\alpha$ and get that

$$\Delta(p, x, y, \tilde{y}) > (1 - \alpha) \cdot \Delta(x, y, \tilde{y}) + c_1^2 - 1 \tag{33}$$

$$= (1 - \alpha) \cdot \Delta(x, y, \tilde{y}) - \Delta_{\tilde{\mathcal{D}}} \cdot (1 - \alpha)^2 \tag{34}$$

Since, $\Delta(x, y, \tilde{y})$ is at least $\Delta_{\tilde{\mathcal{D}}}$ we get that:

$$\Delta(p, x, y, \tilde{y}) > (1 - \alpha) \cdot \Delta(x, y, \tilde{y}) - \Delta(x, y, \tilde{y}) \cdot (1 - \alpha)^2 \tag{35}$$

And Equation 32 follow from algebraic manipulations. $\qquad\square$

