# OpenReview forum: "The Learnability of In-Context Learning"
_NeurIPS.cc/2023/Conference — NeurIPS 2023 poster_

### Official Review · Reviewer_86vc · 2023-07-06

**Soundness:** 3 good
**Presentation:** 3 good
**Contribution:** 3 good
**Rating:** 6
**Confidence:** 3

**Summary:**

The authors propose a PAC framework to analyze the expressiveness power of in-context learning in a finite sample complexity scheme. The framework consists of largely two parts, first is the initial pretraining of the next token prediction phase, and the second is the in-context learning phase. Regarding the pertaining data distribution as a mixture of latent tasks. authors provide an understanding that polynomial sample complexity guarantees in-context learnability.

**Strengths:**

- Define a PAC framework to investigate in-context learning
- Reduce the sample complexity needed for in-context learning to polynomial complexity, which is confined to infinite samples in previous work

**Weaknesses:**

- The number of the model parameter is also a crucial factor for in-context learning, but it seems that there is not enough handling for model complexity.

**Questions:**

- In line 143, the authors say they aim to provide an analysis of in-context learnability of “large model”. With the specified sample complexity, what is the guideline for that “large model”?

**Limitations:**

- Lack of experimental evidence to support the theory (at least experiments on synthetic data would be helpful)

---

> ### Author Rebuttal · Authors · 2023-08-08
>
> We thank you for your thoughtful and supportive feedback.
>
> 1. Regarding the model size, intuitively the learning algorithms in Assumption 1 is a large language model that fits the pre-training distribution well enough. By assuming a scaling law behavior with respect to the model size, one might get an emergent phenomenon in which the capability of recognizing certain downstream tasks with in-context learning is emergent from a certain model size which depends on the margin of the downstream task. We will add a discussion on this avenue for expansion in the camera read version.
>
> 2. Following your request, we promptly conducted an experiment to support our theory. In particular, the analyzed setting includes as a special case the mixture of HMMs examined in [1]. Therefore, the GINC simulations already conducted in Section 4 of their paper lend support to our theory. Moreover, as shown in the attached figure, our PAC bounds provide finite sample complexity, unlike the asymptotic analysis in [2]. We were thus able to apply these bounds to the GINC dataset. This demonstrates that the analyzed framework predicts the in-context capabilities of pre-trained LLMs.
>
> 3. Regarding guidelines for large language models (LLMs), our analysis reveals that the accuracy of in-context learning for large language models (LLMs) is determined by three key factors. First, the ability to recognize relevant tasks among those learned during pretraining. Second, the ability to overcome distribution drift caused by the unnatural concatenation of in-context examples. And third, the performance of the LLM once it identifies the right task. Importantly, our bounds suggest that as the number of in-context examples increases, improvements from the first two factors plateau (Theorem 1). Thus, for further gains, one must utilize a better pretrained LLM (Theorem 2 proof). We will discuss these implications of our theorems in more detail in the camera-ready version.
>
> [1] Xie, Sang Michael, et al.  “An explanation of in-context 541 learning as implicit Bayesian inference.” ICLR 2022.

---

### Official Review · Reviewer_xPBs · 2023-07-06

**Soundness:** 3 good
**Presentation:** 3 good
**Contribution:** 3 good
**Rating:** 6
**Confidence:** 3

**Summary:**


This paper attempts to formalise the few-shot in-context learning phenomenon observed in large language models. To that end, they make a set of assumptions about the underlying data-generating distributions, pretrained models, etc and try to formalise the task of in-context learning in the PAC framework.

The main empirical phenomenon that is the subject of study is the ability of LLMs to predict a query input accurately after receiving a sequence of pairs of inputs and labels. Broadly, they prove that if the distributions over strings for pretraining and providing in-context examples are defined in a certain way and satisfy a set of assumptions, then for large enough k, f(y | x_1, y_1, …, x_k, y_k) will be approximately correct with high probability (in the PAC sense) where f is the language model.


In their framework, they assume that the target task during in-context learning is part of the pretraining distribution based on prior observations in empirical works. The hypothesis class for pretraining is a set of mixture distributions of downstream distributions. The downstream distributions can be seen as a distribution over pairs of inputs and labels. The sampling from the pretraining distribution can be decomposed into sampling a task from a prior distribution over the set of tasks and then sampling from the respective downstream distribution corresponding to the task. They assume that we have an accurate language model (in the PAC sense) where the error with the target distribution is bounded in terms of TVD of the conditional next word distribution -- based on the fact that we have LMs that are accurate in modelling the underlying distribution. Given such an accurate probabilistic LM and inputs from such distribution, they show that given a prompt, for large enough k, the error for f(y | x_1, y_1, …, x_k, y_k) with respect to the Bayes optimal predictor is less than ε with high probability.



**Strengths:**


 (S1) I think the formalisation is interesting to some extent, and their analysis provides some intuition as to why few-shot learning, as observed in GPT-3, works to some degree. I think the key idea is to use the fact that the ratio P_A(p) and P_B(p) converge to zero as k gets larger, where B is the target task, and A is a different task. This helps distinguish the tasks and improves the margin between the correct label and incorrect label for the LM as the size of the prompt goes larger.

Another interesting part about their framework is that they treat it as a distribution modelling task, unlike some other recent efforts to formalise in-context learning [2, 3] and derive sample complexities [1].

They have taken a more physics-like approach, where they have assumed and defined the models and data-generating distributions in a certain way based on empirical observations and tried to work out why prompts could lead to correct answers if the pretraining data was not of that form. The four assumptions are not completely unrealistic. At least within their framework, it is somewhat clear why flipping some labels is not as detrimental as one would expect.

(S2)  In-context learning has been an intriguing phenomenon, with multiple works seeking to understand it. I think this paper takes a step towards that and could be useful to other researchers working in this area.

(S3) The paper is well written, easy to follow, and the arguments are clearly presented.



**Weaknesses:**

While the analysis and the framework are interesting, I think there are a few issues with the framework that limits its applicability in helping us understand in-context learning.

(W1) The way the pretraining distribution is defined is not necessarily reflective of the real-world data. To be clear, it need not always be and depends on the nature of the theoretical work. For instance, the setup [1-3] is quite simplified, but they still allow us to train Transformers on those specific tasks and test how well the theory predicts the sample complexities in the simplified setting. In the case of this paper, the theory seems to be an attempt to directly model in-context learning in the real-world scenario, and the way the pretraining distribution is defined seems a bit detached from real-world text.

Additionally, if my understanding is correct (correct me if I am wrong), each example from the pretraining distribution seems to be an input example followed by a label from a downstream task. This seems a bit far from the way LLMs are trained as well. To some extent, it seems like the pretraining and downstream distributions are designed to satisfy some properties favourable for the end result but not necessarily reflective of the real data. I understand that it is not possible to precisely define real-world data, but given the goal of the paper, it seems like the value of results does depend on how well the data-generating distribution reflects real-world data.

(W2) It seems like the sample complexities are not necessarily predictive of the number of few-shot examples needed for in-context learning. I think it stems from the way the framework is defined and the disconnect with practice. Standard learning theoretic frameworks such as PAC models are entirely formal abstractions and do not contain assumptions about the real world (apart from train and test being the same distribution) -- hence allowing worst-case sample complexities to be meaningful. It also allows us to analyse new algorithms and learning problems. In this scenario, however, it is difficult to see how this framework can be used to analyse the problem or new algorithms further. The framework used in [1] is simplified but is still within a setting where models can be trained and tested to evaluate how well the sample complexities reflect the true performance within the simplified setting.

[1] Transformers as Algorithms: Generalization and Stability in In-context Learning
[2] What Can Transformers Learn In-Context? A Case Study of Simple Function Classes
[3] Transformers Learn In-Context by Gradient Descent. 2022

-------------------------------------------
Typos:
L266: Section ??
L270: That it -> That is
L305: Section ??

I have given a score of 6 for now, but I am open to changing my score based on the authors' responses and discussions with other reviewers.

**Questions:**

NA

---

> ### Author Rebuttal · Authors · 2023-08-08
>
> We thank you for your thoughtful and supportive feedback.
>
> 1. We agree that in real world data there is an additional distribution shift which is caused by the fact that tasks in the pre-training mixture distribution usually use more soft labels and more flexible input formats. That being said, to the best of our knowledge the analyzed data distribution is still the most realistic one that has been analyzed so far. For example, the analyzed setting includes as a special case the mixture of HMMs that is analyzed in [2]. Moreover, note that the underlying mechanism behind our results is Lemma 1, which can be modified easily for other input formats. Hence, one needs to assume that input examples are followed by a label from downstream tasks, only for bounding the zero-one loss, but soft in-context learning as defined in [1] is guaranteed also with soft inputs formats.
>
> 2. Furthermore, our framework allows for an empirical evaluation of how accurately the sample complexities reflect true performance within the rather realistic simplified setting. In particular, since the analyzed setting includes as a special case the mixture of HMMs examined in [2], we can apply our bound to the GINC simulations they conducted in Section 4 of their paper, as shown in the attached figure our bounds correlate well with the trends of real world in-context learning performance, even though they deviate significantly from the exact accuracies. Notably, the analysis in [2] is only asymptotic and does not provide finite sample complexity guarantees. Additionally, our theoretical definitions permit analyzing novel in-context learning algorithms such as CSCGs [3].
>
> #### Details regarding our guarantees for the GINC dataset
>
> The estimation of the KL divergence between the different HMM components significantly influence our bounds and is somewhat unreliable, since the GINC dataset from [2] violates their Assumption 5 and assign zeros probabilities to some strings. To overcome this issue, we cliped the lowest probabilities to $10^{-32}$, and during the Monte Carlo estimation of the KL divergence we omitted samples that violate this lower bound. As a result, we estimate that the KL divergence between the different components to be approximately $15$ and substitute this value into our bounds. For reproducibility, we provide an implementation of our bounds below.
>
> ```python
> def our_bound(kl = 15., delta = .99, lowest_prob=1e-32, task_length=5, mixture_size=5):
>     accuracy = np.linspace(1, 99.9, 100)
>     epsilon = 1 - (accuracy / 100)
>     a = -16 * np.log(delta) * (np.log(lowest_prob) ** 2) / (kl ** 2)
>     b = -2 * np.log(2 * epsilon / mixture_size) / (kl + (np.log(lowest_prob) / task_length))
>     return np.maximum(a, b), accuracy
> ```
>
> ### References:
>
> [1] Olsson, Catherine, et al. "In-context learning and induction heads." arXiv preprint arXiv:2209.11895 (2022).
>
> [2] Xie, Sang Michael, et al.  “An explanation of in-context 541 learning as implicit Bayesian inference.” ICLR 2022.
>
> [3] Swaminathan, Sivaramakrishnan, et al. "Schema-learning and rebinding as mechanisms of in-context learning and emergence." arXiv preprint arXiv:2307.01201 (2023).‏

---

### Official Review · Reviewer_42uY · 2023-07-09

**Soundness:** 3 good
**Presentation:** 2 fair
**Contribution:** 3 good
**Rating:** 6
**Confidence:** 3

**Summary:**

The paper presents a theoretical framework for in-context learnability. The framework is grounded in the Probably Approximately Correct (PAC) learning theory and provides the first-ever finite sample complexity results for the in-context learning setup. The authors' approach involves a pretraining phase followed by an in-context learning phase where the training examples of the downstream task are concatenated in the input. The paper argues that, under specific conditions, latent tasks in the pretraining distribution can be effectively learned via in-context learning without changing the model's weights, even when the input significantly deviates from the pretraining distribution. This finding aligns with recent empirical results and suggests that in-context learning is more about task identification than learning.


**Strengths:**

- The paper is well-written and clearly presented;
- The paper tackles an important and underexplored theoretical aspect of in-context learning in LLMs.
- It presents the first-of-its-kind PAC-based framework for in-context learnability, which could pave the way for further theoretical investigations.
. The paper links its theoretical analysis to recent empirical findings, providing a sound validation for its results.


**Weaknesses:**

- While the paper discusses a pretraining phase and an in-context learning phase, it doesn't clearly address how the transition between these phases is managed or optimized. Also, the instruction-tuning phase might be missing to ensure the success of in-context learning and mixture of tasks as defined in the paper;
- Some empirical valuations including flipping labels could also be valuable to add [1, 2];


[1] Min, Sewon, et al. "Metaicl: Learning to learn in context." arXiv preprint arXiv:2110.15943 (2021).

[2] Wei, Jerry, et al. "Symbol tuning improves in-context learning in language models." arXiv preprint arXiv:2305.08298 (2023).




**Questions:**

- What if the downstream task is OOD from the pretraining data (which is common to see in terms of the generalization ability of instruction-tuned large language models), will the assumption, and conclusion still hold in Lemma 1?


**Limitations:**

- The PAC framework employed may not cover all aspects or scenarios of in-context learning with LLMs (order, variance in prompts, chain-of-thought prompting, generalization, emergent abilities, and so on).
- Some inconsistencies in line 266, 305 with the appendix could be improved.

---

> ### Author Rebuttal · Authors · 2023-08-08
>
> We thank you for your thoughtful and supportive feedback.
>
> 1. Regarding the transition between the pretraining phase and the in-context learning phase, please note that we analyzed vanilla in-context few-shot learning as introduced in the GPT-3 paper [1], which does not have an instruction-tuning phase between the two phases. That said, our Definition 1 of in-context learning is general enough to potentially incorporate an instruction-tuning phase. Analyzing the effect of such a phase is an exciting open problem that we leave for future work.
>
> 2. Regarding the ability to generalize to out-of-distribution downstream tasks, our results do not fully explain the capability of learning new tasks not included in the pre-training distribution, as acknowledged in the conclusion. However, since our results also hold under weaker approximate independence assumptions (see our response to o2CM), they cover the ability of large language models to infer latent learning algorithms like linear regression from a mixture of algorithms learned during pre-training . Importantly, inferring the right learning algorithm among already learned algorithms might provide generalization to out-of-distribution downstream tasks. Please see our answer to yYWi for more details about the ability to infer latent learning algorithms.
>
> 3. Regarding variance and order variation see our first answer to o2CM.
>
> 4. Regarding emergent abilities, see our answer to 86vc.
>
> [1] Brown, Tom, et al. "Language models are few-shot learners." NeurIPS 2020.

---

### Official Review · Reviewer_yYWi · 2023-07-12

**Soundness:** 3 good
**Presentation:** 3 good
**Contribution:** 3 good
**Rating:** 6
**Confidence:** 3

**Summary:**

This paper studies the PAC-learnability of in-context learning when the pretraining distribution is a mixture of latent tasks, and the downstream task belongs to one of them. In addition to this mixure-of-tasks assumption, the other non-trivial assumptions the authors make include: the pretrained model can approximate the pretraining distribution arbitrarily close with a polynomial number of examples, and that the adjacent strings to be concatenated are approximately independent. Based on these assumptions, the author provides sample complexity results to demonstrate in-context learnability. Particularly, the correctness of the downstream task labels does not affect the validity of the conclusions.

**Strengths:**

* The problem addressed in this paper, namely theoretical understanding of in-context learning, is undoubtedly important and timely.
* The author addresses the PAC-learnability of in-context learning, which has not been well-studied. The problem formulation and results proposed by the authors are also interesting and novel.
* The overall presentation is clear, with clear motivations and summaries for each part (although several details need to be improved, see Weaknesses).

**Weaknesses:**

* The validity of the overarching mixture-of-tasks assumption for the pretraining distribution may need more justification. It appears to oversimplify the problem given the messy pretraining corpora and the novel downstream tasks used in practice (although it generalizes the similar assumption made by Xie et al. 2022, this limitation still exists).
* The insensitivity to the correctness of in-context examples' labels may contradict recent observations. For instance, Wei et al. 2023 demonstrate that LLMs can do linear regression in in-context learning, which is clearly dependent on labels and thus cannot be directly explained by this paper.
* There are some issues with the current presentation and writing that impede readability, such as:
    - The input and output of function $f_\theta$ should be made clearer. Based on the context, its input seems to be a string, but in equation (4) it has a conditional-style input which is not explained in the main text.
    - The meaning of $x$, $o$, and $s$ should be made clearer, preferably with some examples. The Kleene star symbol should also be clarified.
    - Typos, for instance run-away references to the appendix, Line 297 "other another", Line 362 "works designed" -> "works are designed".

**Questions:**

Is the framework extendable to explain the observation where in-context learning can do label-sensitive linear regression?

**Limitations:**

The authors have not discussed the limitations.

---

> ### Author Rebuttal · Authors · 2023-08-08
>
> We thank you for your thoughtful and supportive feedback.
>
> 1. We agree that pre-training data in the real world is often messy, and we will acknowledge that as a limitation in the camera-ready paper. Note that we already discussed some limitations of our work in the last paragraph of Section 5. Specifically, beyond the capability of learning new tasks, our framework also lacks a formal explanation of the connection between model size and the efficiency of in-context learning.
>
> 2. Regarding the ability to perform label-sensitive linear regression, our results do not fully explain the capability of learning new tasks not included in the pre-training distribution, as acknowledged in the conclusion. However, since our results also hold under weaker approximate independence assumptions (see our response to o2CM), they cover the ability of large language models to infer latent learning algorithms like linear regression from a mixture of algorithms learned during pre-training. Importantly, when the different mixture components are learning algorithms, the labels become the dominant contributors to the Kullback-Leibler divergence between components, rendering Theorem 2 ineffective. In such cases, one could prove a similar theory relying solely on correct labels, resulting in more relaxed conditions than the conditions in line 252 ($\Delta_{\text{KL}}>8\log\frac{1}{c_{1}\cdot c_{2}}$). We will clarify the distinction between label-sensitive and label-insensitive results in the camera-ready version. Finally, when a mixture component is a learning algorithm, it is reasonable to assume the appearance of the delimiter token after labels does not constitute a distribution drift from pre-training. Thus, together with the weaker independence assumptions, one could potentially eliminate the condition in line 252 completely.
>
> 3. We apologize for the unclear notation. We will make the notation cleaner for the camera version.
>
>     *  As you correctly inferred $f_\theta$ is a probabilistic model that gets strings as its inputs, and output the likelihood of the input string. Regarding, Equation 4, we apologize  for the abuse of notation, the intended meaning of the conditional notation is $f_{\theta}\left(o_{T}\,|\,o_{1}\dots o_{T-1}\right)\coloneqq\frac{f_{\theta}\left(o_{1}\dots o_{T-1}o_{T}\right)}{f_{\theta}\left(o_{1}\dots o_{T-1}\right)}$.
>     * The meaning of $x$ is the task inputs (see line 99). For example, “What is the name of the first president of the United States?”.
>
>     * The meaning of $o$ is a single token (see line 182). For example, “What”.
>
>     * The kenner star in $\Sigma^{\star}$ stands for the set of sequences over the alphabet $\Sigma$.
>
>     * The meaning of $s$ is a string in $\Sigma^{\star}$  (see line 206).

---

### Official Review · Reviewer_xztc · 2023-07-16

**Soundness:** 2 fair
**Presentation:** 3 good
**Contribution:** 2 fair
**Rating:** 6
**Confidence:** 3

**Summary:**

This paper aims to explain the learnability of in-context learning. The main idea is that the pretraining tasks learn multiple downstream tasks and the prompt specify a particular task.

**Strengths:**

1. The paper addresses the learnability of the in-context learning. It's a hot topic and very few theories are constructed from my knowledge.
2. The authors made several assumptions on in context learning and explain why they make the assumptions clearly.

**Weaknesses:**

Though I am positive towards the paper, I think the main weakness is that the theoretical framework is too far away from the practice. The paper tells a good story but I am not sure what kind of guidance can the theory provide for the in-context learning in practice.

For example, it looks that if we provide larger number of examples in prompts, the bound may be tighter. Is this really true for in-context learning? The authors can totally claim that there is computation limitation so we cannot make the number of examples to be as large as possible in practice. But if so, we also do not need in-context learning. The authors kind of explain the mystery of in-context learning. But kind of not. If we can make the number of examples infinity, supervised learning should also work well. I think the gap here is that the authors do not explain why this kind of in-context learning is better than supervised learning.

Overall, I think the contribution is interesting but not significant.

Minor:
There are some ??'s in the paper. The authors probably want to fix them.

**Questions:**

See the weakness section.

**Limitations:**

This is a theory paper. Societal impact is irrelevant.

---

> ### Author Rebuttal · Authors · 2023-08-08
>
> We thank you for your thoughtful feedback.
>
> 1. Increasing the number of in-context examples has been shown to be beneficial in practice. Evidence for this can be traced back to the GPT-3 paper [1], in which Figure 1.2 clearly demonstrates that performance improves as the number of in-context examples increases. Moreover, these observations have inspired works such as [2] to invent techniques to overcome the computational limitations of using a large number of in-context examples, their figure 1 clearly shows this improvement trend.
>
> 2. Your thoughtful feedback has inspired us to prove a simple corollary of Theorem 2, establishing a case where the sample complexity of in-context learning is provably better than that of supervised learning. Specifically, the downstream tasks will involve learning parity functions over unknown subsets of input bits. To reflect that concatenating independent examples is unnatural in pre-training, the pre-training distribution will mix two distributions corresponding to the downstream parity functions. Importantly, new line tokens will usually not follow labels, and successive examples in pre-training will usually share inputs rather than be independent. The VC dimension of this hypothesis class scales linearly with the number of bits, so supervised learning's sample complexity does too. In contrast, since pre-training reduces the hypothesis class to two relevant parity functions, Theorem 2 gives a constant sample complexity regardless of the number of bits. Thus, this is effectively **few-shot** learning. In essence, the corollary shows pre-training followed by in-context learning can significantly reduce sample complexity compared to supervised learning. This demonstrated advantage is notably stronger than existing results like [3] on pre-training benefits for supervised fine-tuning. Please find below a proof sketch for this corollary.
>
> ### Proof sketch for a corollary of theorem 2 on the advantage of in-context learning over supervised learning:
>
> To utilize Theorem 2, we employ Algorithm 2 from [4] as our pretraining algorithm. Importantly, Theorem 2 from that paper provides strong formal guarantees, ensuring Assumption 1 of our paper holds. Now, since we allow successive examples in pretraining to be independent with low probability $c << 1$, successive examples are approximately independent given the new line token. Therefore, Assumption 2 also holds. Similarly, we will allow incorrect labels and spontaneous new line tokens to occur with the small probability of $c$. This ensures Assumption 3 holds. To overcome distribution drift, we repeat the parity label one hundred times. This results in the Kullback-Leibler divergence between distributions being exactly $\frac{100}{2}\cdot\log\frac{1-2c}{c}$. Consequently, for small enough $c$, the divergence is greater than $8\cdot\log\frac{1}{c\cdot\left(1-c\right)}$. Overall, we have satisfied all conditions for Theorem 2. Therefore, we can conclude that parity functions over unknown input bit subsets are in-context learnable, with sample complexity independent of the number of bits. Rather, the sample complexity depends only on $c$, which determines the similarity between pretraining and downstream tasks. In contrast, the VC dimension of this hypothesis class scales linearly with the number of bits. Consequently, the sample complexity of supervised learning scales linearly with the number of bits, thus completing our proof.
>
> ### References:
>
> [1] Brown, Tom, et al. "Language models are few-shot learners." NeurIPS 2020.
>
> [2] Ratner, Nir, et al. "Parallel Context Windows for Large Language Models." ACL 2023.
>
> [3] Ge, Jiawei, et al. "On the provable advantage of unsupervised pretraining." arXiv preprint arXiv:2303.01566 (2023).
>
> [4] Mahajan, Gaurav, et al. "Learning Hidden Markov Models Using Conditional Samples." COLT  2023.

---

### Official Review · Reviewer_o2CM · 2023-07-29

**Soundness:** 2 fair
**Presentation:** 3 good
**Contribution:** 3 good
**Rating:** 5
**Confidence:** 3

**Summary:**

In-Context Learning (ICL) allows large language models (LLMs) to be easily specialized to natural language downstream tasks. When users input a concatenated string of examples of a particular downstream task, modern LLMs often perform successfully without changing their weights, providing an effective new angle to tackle multiple NLP tasks immediately without fine-tuning.

This paper defines ICL within the PAC learning framework. The authors find that pretrained LLMs learn a mixture distribution of downstream tasks (though these models maximize the likelihood of self-supervised next tokens or masked tokens on the training corpus). Under the mild assumptions, they show that ICL is provably guaranteed for LLMs to uncover the latent task, improving their performance without modifying any weights.

**Strengths:**

The papers accept the crucial observations while providing theoretical guarantees.

-	Even if pre-training is assumed to be a process of learning mixture distribution of downstream tasks, it could never be equivalent to fine-tuning on a target task due to other irrelevant tasks.

-	Providing concatenation of independent examples is not natural for pretrained LLMs because they have never encountered such examples while pre-training.

**Weaknesses:**

While some assumptions are claimed mild for learnability context, there are other assumptions which could not be easily acceptable given the practical behaviors of LLMs.

-	To make the provable bounds practically useful, both $c_1$ must be close to 1. However, the two strings delimited by the newline are not necessarily on paragraph levels. See the more details on the questions.

-	If we increase the size of vocabulary, it would be easier to guarantee the existence of positive $c_2$, but there would be a computational bottleneck while $c_2$ could be still very low, making the bound less practical. See the more details on the questions.

**Questions:**

There are two questions mainly for the Assumption 2 and 3.


(Regarding the Assumption 2)

It is true that we can always find a positive c_1 that can easily bound the fraction in Equation (5) given $s_1$ and $s_2$. One concern is that users of LLM often use the newline character either trying to distinguish different shots of examples or differentiate consecutive paragraphs.

-	For the former case, would it be better to bound min and max of the fraction provided with two different orders $(s_1, \n, s_2)$ and $(s_2, \n, s_1)$? This is because the order of few-shot examples actually matter for the ICL performance.

-	For the latter case, it is difficult to say that two consecutive paragraphs would be relatively independent. It could be independent in a particular task, but not generally in most NLP tasks. For example, Coreference resolution is barely runnable without the previous paragraphs.

In addition, it seems $c_1$ is defined as a strongly uniform constant (rather than depending on the choice of $s_1$ and $s_2$, and even including any possible NLP task). Would it be too strong? All in all, $c_1$ is likely to be extremely small number, thus making the bound less practical.


(Regarding the Assumption 3)

If the size of vocabulary is very large (typically 50k tokens), it becomes easier to find such $c_2$ (likely another uniform constant), but the $c_2$ would be very small yet requiring computational costs to handle large vocabulary. If the size of vocabulary is not large, it becomes harder to guarantee the existence of such $c_2$. Overall, $c_2$ would likely become a tiny positive number though the assumption would hold.


(Regarding the Lemma 1)

How to justify the last line of Lemma 1, where $m_{\hat{D}}$ can be chosen to be polynomial in both $\log \frac{1}{c_1 \cdot c_2}$ and $\frac{1}{\Delta_{KL}}$ within the context of Equation in Line 252? If so, would be the polynomial time both lower- and upper-bounded by the $\log \frac{1}{c_1 \cdot c_2}$?


(Minor questions)

-	Section reference in Line 266 is broken.

**Limitations:**

No specific points are described or probed.

---

> ### Author Rebuttal · Authors · 2023-08-08
>
> We thank you for your thoughtful feedback.
>
> ### Regarding Assumption 2
> Our PAC-style guarantees are worst-case in nature and do not depend on the sampled $s_1$ and $s_2$. That being said, your suggestion is a promising direction for extending our framework to input-dependent bounds, which will elucidate the sensitivity of few-shot in-context learning to the order of the few-shot examples, as well as other input specific characteristics. We will add a discussion on this avenue for expansion in the camera read version.
>
> In addition, your comment encouraged us to tighten the analysis in Theorem 1. We were able to relax the requirement that $4\cdot(1-c_1^2) < \Delta_{\tilde{\mathcal{D}}}$ into the weaker requirement that $1-c_{1}^2<\Delta_{\tilde{\mathcal{D}}}$. Please see the updated proof below. Note that with this tighter analysis our bounds are meaningful for any $c_1>0$, and the effect of $c_1$ on the bounds is only logarithmic.
>
> Furthermore, note that our approximate independence assumptions are used merely for simplicity. In fact, the results remain valid even when we relax the approximate independence assumption to a weaker one, namely that the log-ratio of likelihoods according to different tasks is a martingale. Importantly, this weaker assumption will allow the propagation of information essential for coreference resolution, albeit at the cost of clarity. Please find below a proof sketch for Theorem 1, which relies solely on the relaxed assumption. This proof will be added to the camera-ready version.
>
> ### Regarding the question about Assumption 3
> We agree that this assumption becomes less restrictive as the vocabulary grows larger. Please note that while in practice $c_2$ may be relatively small, the effect of $c_2$ on our bounds is only logarithmic. Therefore, even a small $c_2$ will not significantly impact the bounds.
>
> ### Regarding Lemma 1
> We apologize for our mistake in line 256, it is not the correct expression, the exact form of the sample complexity in Lemma 1 is the maximum between $\frac{\left(\log\frac{1}{\delta}\right)\left(16T^{2}\right)\left(\log^{2}\frac{1}{c_{2}}\right)}{\left(\bigtriangleup_{\text{KL}}\right)^{2}}$ and $\frac{2\log\frac{1}{\epsilon}}{\bigtriangleup_{\text{KL}}-8\log\frac{1}{c_{1}c_{2}}}$  (see line 20 in the supplementary materials). As you can see, the exact form of the sample complexity in Lemma 1 does not contradict the equation on line 252, and these equations are consistent with each other.
>
> ### Updated proof with the approximate independence assumption
> Let $\alpha\coloneqq1-\sqrt{\frac{1-c_{1}^{2}}{\Delta_{\tilde{\mathcal{D}}}}}$. Since we assume that $\Delta_{\tilde{\mathcal{D}}}>1-c_{1}^2$, it follows that $0<\alpha \le1$.  Importantly, Theorem 1 guarantees that $\Delta\left(p,x,y,\tilde{y}\right)>\left(1-\alpha\right)\cdot\Delta\left(x,y,\tilde{y}\right)+c_{1}^2-1$ after a minor modification. We just need to adjust the term $\frac{\Delta\left(x,y,\tilde{y}\right)}{5\cdot c_{!}^{-2}\cdot c_{2}^{-T}\cdot c_{3}^{-1}}$ in line 39 of the supplementary materials to $\frac{\Delta\left(x,y,\tilde{y}\right)}{\left(1-\frac{\alpha}{2}\right)^{-1}\cdot c_{!}^{-2}\cdot c_{2}^{-T}\cdot c_{3}^{-1}}$.
>
> Now, we can choose $\Delta_{\text{pretraining}}=\frac{1}{2}\cdot\alpha\cdot\left(1-\alpha\right)\epsilon$. The proof of Theorem 2 follows the same logic as the original proof, except we separate into cases based on whether the margin is at least $\frac{2\cdot\Delta_{\text{pretraining}}}{\alpha\cdot\left(1-\alpha\right)}$. In the first case, Theorem 1 assures us that for large enough $k$ the ground truth in-context predictor also has margin $\Delta\left(p,x,y,\tilde{y}\right)$ that is greater than $\left(1-\alpha\right)\cdot\Delta\left(x,y,\tilde{y}\right)+c_{1}^{2}-1$.
> Now since $\Delta\left(x,y,\tilde{y}\right)\ge\Delta_{\tilde{\mathcal{D}}}$ and since $\frac{1-c_{1}^{2}}{\Delta_{\tilde{\mathcal{D}}}}=\left(1-\alpha\right)^{2}$, we have that $\Delta\left(p,x,y,\tilde{y}\right)>\Delta\left(x,y,\tilde{y}\right)\left(1-\alpha\right)\cdot\alpha$. From here the proof remains unchanged.
>
> ### Proof sketch with the relaxed assumption
> Our goal is to find an alternative to Hoeffding's inequality in Equation 8 of the supplementary materials. To accomplish this, we will replace the approximate independence assumption with a weaker one that still enables a concentration inequality. Specifically, we will assume the following sequence of random variables forms a submartingale:
> $\log\frac{\mathbb{P_{\phi^{\star}}}\left(p\right)}{\mathbb{P_{\phi}}\left(p\right)}-\left|p\right|\cdot\text{KL}\left(\mathbb{P_{\phi^{\star}}},\mathbb{P_{\phi}}\right)$
> where $p$ is a concatenation of $n$ in-context examples. With this more relaxed assumption, we can utilize the Azuma inequality rather than Hoeffding's inequality, thus avoiding the approximate independence assumption. Importantly, this new assumption implicitly includes Assumption 3, as it handles the distribution drift caused by the artificial new line token in an implicit way, rather than the explicit treatment we currently have in Equation 2 of the supplementary materials.
>
> So far, we have successfully adapted the proof of Lemma 1 to incorporate the relaxed assumption. To prove Theorem 1 under this relaxed assumption as well, we must quantify the efficacy of the Bayes optimal classifier that classifies inputs based on the ground truth **prompted** mixture component. For simplicity, we will assume that this classifier performs at least as well as the Bayes optimal classifier that classifies inputs according to the **unprompted** ground truth mixture component. Note that this is a natural assumption when the downstream task is a learning algorithm such as linear regression. Finally, note that with the approximate independence assumption, predictions made using the prompted ground truth mixture component can be associated with those of the unprompted ground truth mixture component, rendering this assumption unnecessary.

---

> > ### Comment · Reviewer_o2CM · 2023-08-18
> > **Thanks for your rebuttal.**
> >
> > Thanks the authors for their additional work to clarify my comments and feedback. Indeed this work will be better shine by addressing some of your extension and tighter bounds with respect to my suggestions. Hope these will be included in the final draft and future directions. Upon this feedback, I have increased my score.

---

> > > ### Author Response · Authors · 2023-08-20
> > > **Thanks for your response**
> > >
> > > We thank the reviewer for helping us improve our paper. We assure you that we will include the extension and tighter bounds in the camera-ready version of the paper. Would you consider increasing the Soundness score for our paper as well?

---

### Author Rebuttal · Authors · 2023-08-08

We thank all the reviewers for their thoughtful feedback. We apologize for the broken links to the appendix. The link in line 266 should point to section 1 in the appendix, while the link in line 305 should point to section 2 in the appendix.

---

### Decision · Program_Chairs · 2023-09-21

**Decision:**

Accept (poster)

**Comment:**

This paper proposes a theoretical framework for understanding in-context learning through the lens of PAC-learning. Under the assumption that the pre-training distribution contains a mixture of latent tasks, the authors derive a polynomial sample-complexity bound for learning to perform these tasks in-context.

The consensus is quite positive on this paper, and all of the reviewers agree that this is a useful contribution to understanding in-context learning. The overarching theme in the concerns is that the theory relies on assumptions which may not always reflect the practice.

However, the positive outweigh the negative, especially given that the authors acknowledge these limitations in the paper. I recommend acceptance.